# Designing Component Interfaces for the Circular Economy—A Case Study for Product-As-A-Service Business Models in the Automotive Industry

Philip Krummeck [1], Yagmur Damla Dokur [2,*], Daniel Braun [1], Steffen Kiemel [2] and Robert Miehe [2]

[1] Mercedes-Benz AG, Leibnizstr. 2, 71032 Boeblingen, Germany
[2] Fraunhofer Institute for Manufacturing Engineering and Automation IPA, Nobelstrasse 12, 70569 Stuttgart, Germany
* Correspondence: yagmur.damla.dokur@ipa.fraunhofer.de; Tel.: +49-711-970-1403

**Abstract:** The resource-intensive automotive industry offers great potential to avoid waste through new circular business models. However, these new business models require technical innovations that enable the rapid dismantling of add-on parts. In this paper, we design new mechanical interfaces that enable fast and non-destructive dismantling while still fulfilling all technical requirements and develop a general model for the evaluation of disassembly capability. For this purpose, the current dismantling options of add-on parts are first examined and evaluated concerning defined KPIs using the example of the front bumper. Based on the analysis, the requirements as well as various solution principles for the new interface concept can be derived. The necessity of removing neighboring components is identified as the main challenge for rapid dismantling. Two different concepts for the interfaces were developed by inserting an intermediate level as a connecting part between the front bumper and the front module. We prove that by redesigning and reconstructing the interfaces the number of process steps required to remove the front bumper could be reduced by roughly 60% compared to current interface solutions. The developed methodology should be applied to other components of a vehicle to create a greater positive environmental, economic and societal impact.

**Keywords:** circular economy; circular business models; life cycle engineering; automotive industry; sustainability; design for recycling

## 1. Introduction

The global increase in demand for consumer goods, the shortage of raw materials, and, at the same time, more restrictive legal $CO_2$ and recycling regulations are forcing many companies to rethink their current business models [1]. By announcing the "Green Deal" in 2019, the EU has developed a growth strategy, which aims to release no net greenhouse gas emissions by 2050 [2]. In many industrial sectors, solutions are therefore being sought to counteract the aforementioned problems [3]. A promising approach to reach these targets is the transformation of the current linear economy. In the linear economy, raw materials are processed into goods; goods are sold to the customer and finally disposed of when the product is no longer functioning or needed. This type of economy assumes unlimited resources and results in increased resource waste in the long run [4–6]. Hence, the circular economy offers a solution to decouple economic growth from the consumption of primary resources [4,5]. The transition to a circular economy needs to be implemented rapidly given the significant changes in our natural environment caused by the linear economy [7]. A further important driver for the transformation to a circular economy is represented by the fact that companies will increasingly incur costs for the use of the environment. Taking these costs into account in operational cost accounting makes circular concepts more and more attractive [8].

The automotive industry is already undergoing a significant transformation and has strongly shaped the image of cities and urban areas in recent decades. The number of 1.2 billion motor vehicles in the world today, and a predicted two billion in 2035, illustrate the importance and influence of this product in people's lives [9]. As the number of new vehicles rises, the number of end-of-life vehicles will also increase in the long term [10–12]. Although many manufacturers have already designed their vehicles in part for reuse and recycling, according to the Federal Environment Agency, vehicles from the premium segment of the Mercedes-Benz, BMW, and Audi brands in particular were often underrepresented in shredder plants in 2016 concerning the registered vehicles 15 years prior. High demand from abroad and an associated disproportionate export to foreign countries compared to other vehicle manufacturers is given as possible reason for this challenge. Accordingly, the exported vehicles do not end up for recycling in Germany [13].

In 2019, approximately 3.12 million motor vehicles were decommissioned in Germany. A large proportion of the vehicles (80.1% in total) were exported as used vehicles, with 2.16 million vehicles exported to EU countries and 0.34 million vehicles exported to non-EU countries. The vehicles were then re-registered for road use in the respective countries. Only around 460,000 vehicles (14.7%) of the 3.12 million decommissioned vehicles were recycled as end-of-life vehicles in 2019. For a small proportion of 160,000 vehicles (5.2%), the whereabouts could not be clarified [13]. The whereabouts of the permanently decommissioned vehicles in Germany in 2019 are shown graphically in Figure 1.

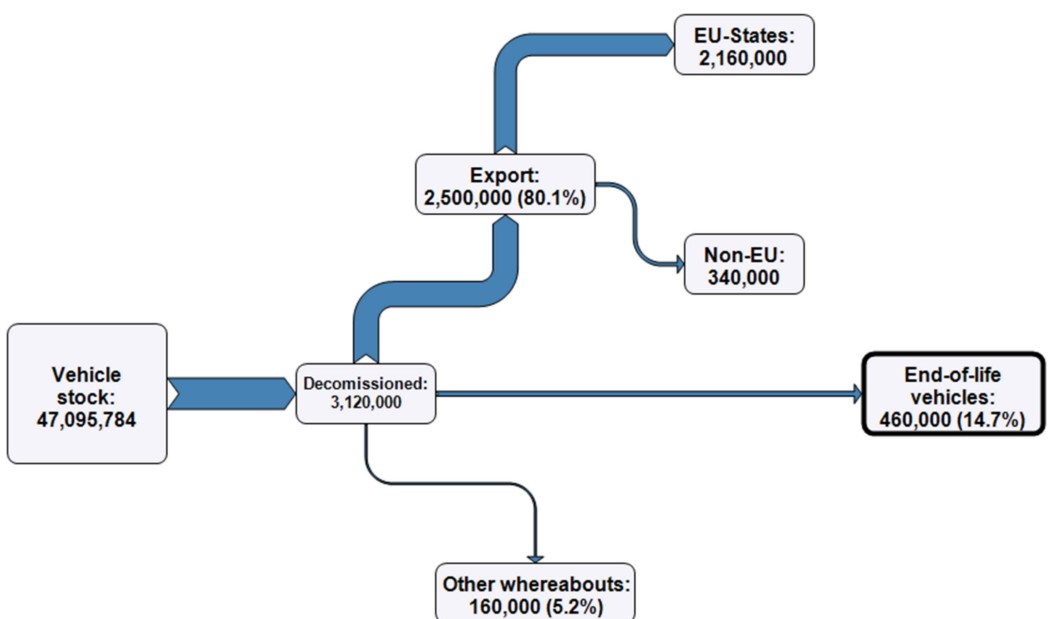

**Figure 1.** Whereabouts of decommissioned vehicles in Germany in 2019, data from [13].

Due to the high export rate of decommissioned cars a large proportion of the materials used within the cars are lost to the local economy [14,15]. Moreover, increasing material criticality issues in industrial enterprises lead to the need for circular solutions [16,17]. To keep vehicles and the resources used in a cycle, a circular vehicle is designed as part of a new business model, in which the company remains the owner of its products and is therefore also responsible for its disposal or recycling [18]. According to Martins et al. [6], new circular business models within the automotive industry also require new vehicles that can be easily repaired and maintained, and easily disassembled at the end of the life cycle to recover the resources and reintroduce them into the material cycle [6]. The main idea of the circular vehicle concept is to keep the vehicle always up to date with the latest technology by replacing only individual components and modules. In this concept, no completely new vehicles are produced, but existing ones are constantly refreshed. However, vehicles have so far been designed for one-off assembly and time-intensive workshop repair. Consequently,

the development of suitable interfaces through which the various components of the vehicle can be easily changed is a crucial prerequisite for the realization of the concept.

However, no solutions have yet been provided for the circular refreshing of add-on parts. For instance, the front bumper of a vehicle has a high replacement rate in the event of an accident [19–22]. On the other hand, the front bumper contributes to a large extent to the front design of the vehicle [23]. In the context of different equipment lines or a model update, this component is often changed and therefore requires rapid disassembly concerning the intended concept of vehicle refreshing. The problem described leads to the research question of this paper: How can component interfaces of add-on parts be conceptualized and designed to enable circular refreshing of vehicles? We systematically worked out how a joining concept of an add-on-part must be designed in order to maintain direct accessibility during assembly/disassembly while minimizing the need to disassemble neighboring components beforehand. The methodology is developed at the example of a front bumper of a vehicle.

## 2. Theoretical Background

The circular economy is mainly described by the principles of reduce, reuse, and recycle [24–28]. At the beginning of the product life cycle, raw materials are mined. After designing and manufacturing subcomponents, the final product is assembled. This is sold to the customer, used by the customer, and returned to the company at the end of the product's life. Subsequently, the product is dismantled, and the raw materials used are recovered. Recovered raw materials are recycled and can be returned to the production process again. Non-recoverable raw materials are disposed of. The removed subcomponents are reprocessed and requalified and will also be brought back into the production process [29]. The concept of the circular economy is shown schematically in Figure 2.

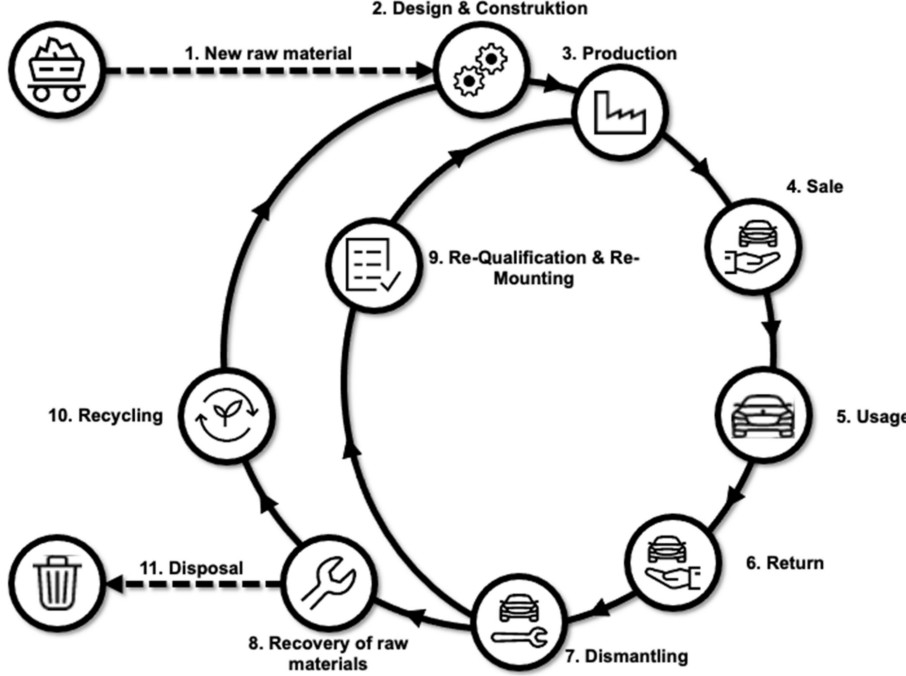

**Figure 2.** The concept of the circular economy adapted from Hjorth and Chrysostomou [29].

However, the circular economy also brings some challenges, especially in terms of technical barriers related to implementation [28,30,31]. Products must be designed to be suitable for dismantling, reuse, and recycling [32]. In addition, not all materials can be reused indefinitely. Contamination of materials can prevent their proper recycling [25]. Vehicles, for instance, contain a variety of materials such as metals, plastics, and glass [9]. To

ensure that these materials are not wasted but reused and recycled, Directive 2000/53/EC of the European Parliament and of the Council of 18 September 2000 on end-of-life vehicles was presented in 2000. This directive stipulates that new vehicles must meet a reuse and/or recycling rate of 85% by weight or a reuse and/or recovery rate of 95% by weight [33]. Reuse is defined in this context as "actions whereby end-of-life vehicle components are used for the same purpose for which they were designed" [33]. Recycling is described as "the reprocessing in a production process of waste quantities for the original purpose or other purposes, but excluding energy recovery" [33]. Remanufacturing is defined as "a process of returning a used product to at least original equipment manufacturer (OEM) performance specification from the customers' perspective and giving the resultant product a warranty that is at least equal to that of a newly manufactured equivalent" [34].

Recent progress in circular automotive manufacturing in the scientific community focuses on battery recycling [35,36]. Glöser-Chahoud et al. [37] investigated the end-of-life (EoL) removal of lithium-ion batteries (LIB) from electric vehicles. Although battery production has been steadily highly automated in recent years, the disassembly of end-of-life batteries is rather simplified. However, manual disassembly is associated with high labor costs. Accordingly, the authors conclude that a systematic highly automated industrial disassembly line, similar to the assembly processes in battery production, is necessary for an economic closed-loop circulation of EoL-LIB from electric vehicles [37]. In a study published in 2016, Diener and Tillman [38] provide a summarizing overview of vehicle end-of-life management. It is pointed out that the realization of recycling has some challenges. For example, the proportion of components replaced during vehicle maintenance can vary widely. The recycling of removed components is limited by low quantities but is also limited by logistics and quality standards for scrap [38]. Hallack et al. [39] developed a systematic Design for Recycling approach in relation to the recycling of plastics in automotive exteriors and investigated challenges, factors and practices. Various challenges for the recycling of plastic components in the exterior are identified. These include the separation of components of an exterior plastic part, access and complexity of the respective components, and a long disassembly time. In addition, there are challenges related to the materials used, such as a large number of different plastics [39]. Parsa and Saadat [40] examine the dismantling of end-of-life vehicles. The authors state that manual disassembly is not economically viable and robotic systems are not robust when it comes to complex disassembly processes. By using human-robot collaboration, complex disassembly processes can be handled by leveraging the flexibility of humans and the repeatability and accuracy of a robot [40].

For a successful realization of the circular economy in the automotive industry, the following points, in particular, have been defined in the state of the art: New business models and technical innovations are needed that are compatible with the circular economy [6]. Moreover, circular business models have to be economically viable [41]. The current time-intensive dismantling of components from vehicles in workshops is not economically viable for industrial dismantling [37]. Therefore, vehicle disassembly needs to take place in an industrial disassembly facility to be cost-effective [37]. In addition, facilitated access to the connection points of the individual components must be ensured [42]. The following chapters describe the process and the criteria used to develop a new interface for the exchange of add-on parts.

## 3. Methods

This work aims to develop a methodology for the systematic analysis and improvement of components with regard to their ease of disassembly. As a first step, a benchmark will be carried out in which the current interface solutions of various vehicle manufacturers will be investigated and evaluated for ease of dismantling according to defined criteria based on the VDI Guideline 2243. This will serve to identify potential weaknesses and strengths of the current interface solutions concerning easy and fast disassembly.

Based on the results of the benchmark, requirements for a new interface concept will be derived to eliminate the identified weaknesses. The result is a list of requirements and forms the basis for the subsequent design and construction process. In the final step, the necessary process steps for dismantling the front bumper of a current solution are compared with the newly designed concept. The procedure is shown in Figure 3.

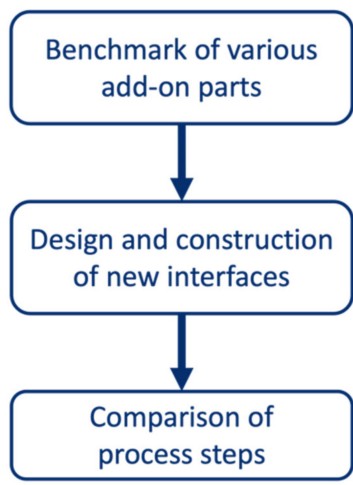

**Figure 3.** The research process of this paper.

For the concept development of new interfaces, the first step is to investigate the required process steps for dismantling the add-on parts at various automobile manufacturers. The analysis aims to identify concepts that are already suitable for the desired goal of the fast and simple dismantling of the add-on parts and which potential weaknesses exist in this respect. To create the benchmark, one representative vehicle from each of the eight different manufacturers is examined. Since the sales market for electric cars is expected to become larger, the analysis only examines vehicles with electric drives from the premium segment, as the front area of these vehicles differs greatly in part from vehicles with internal combustion engines due to the different drive technologies [43]. For example, the function of the radiator grille is omitted in an electric vehicle due to the absence of an internal combustion engine in the front area of the vehicle. In addition, this is to ensure that the vehicles remain comparable. The online tool A2Mac1—Automotive Benchmarking is used, through which various vehicle manufacturers and their models can be accessed. This tool can be used to examine and compare entire vehicle models, assemblies, and individual parts, and it enables the comparison of various add-on parts and their connection to the vehicle. Moreover, the tool can also be applied to determine which other components are adjacent to the add-on parts and also need to be removed during disassembly. In addition, it allows an investigation into the use of materials and the various cable connections.

To evaluate the interface solutions, three criteria (interface type, interface diversity, interface accessibility) from VDI 2243 [44], as well as two self-defined criteria (cable connections, material diversity), are used (see Table 1 below). The assessment by the criterion of the interface type is categorized into non-destructively solvable [ND], partially destructive [PD], and destructive [D] connections. Interface diversity refers to the use of different connection types such as screw connections, snap hooks, and so on. The interface accessibility indicates the number of neighboring components that must additionally be removed to be able to disassemble the desired component. The criterion of cable connections describes the number of electronic connections to be disconnected when disassembling the component. Finally, the material diversity criterion evaluates the number of materials used in the component to be disassembled. The criteria of interface type, diversity, and accessibility, as well as the number of cable connections, make it clear to what extent simple and fast dismantling of add-on parts is possible. The criterion of material diversity refers to the possibility of being able to recycle the concept as easily as possible and without major

material separation. The self-generated rating ranges from five (most suitable) to one (least suitable). At the end of the analysis, the rating scale of each concept is transferred to a common network diagram to obtain an overview of the properties of each concept.

**Table 1.** Evaluation criteria of the interfaces.

| Criteria | Evaluation | | | | |
|---|---|---|---|---|---|
| | **1** | **2** | **3** | **4** | **5** |
| Interface type (non-destructively solvable [ND], partially destructive [PD], destructive [D]) | D | PD + D | ND + D | ND + PD | ND |
| Interface diversity (screws, snap hooks, etc.) | 5 or more | 4 | 3 | 2 | 1 |
| Interface accessibility (disassembly of neighboring components) | 4 or more | 3 | 2 | 1 | 0 |
| Cable connections (sensors, lights, etc.) | 5 or more | 4 | 3 | 2 | 1 |
| Material diversity (different types of plastic) | 5 or more | 4 | 3 | 2 | 1 |

The general procedure of the design process is described in the VDI guideline 2221 and should serve the designer as a guide for a structured design [45]. It is important to mention at this point that a design process is not a rigid sequence of steps. Rather, it is an iterative procedure in which optimization is to be achieved step by step by going back to previous work steps [45]. For recycling-oriented product development, the VDI Guideline 2243 can be regarded as one of the most important documents [44]. This guideline aims to provide the designer with information and decisions for each phase of product development regarding better recyclability of technical products and contains technical recycling criteria such as the accessibility of the installed components, the connection types, and the variety of connections as well as the disassembly time. One design aspect here is the standardization of fasteners. This eliminates the need for tool changes during disassembly and reassembly, which results in a reduced disassembly and reassembly time and thus lower costs [44]. Moreover, it is recommended to use only one type of plastic or compatible plastics for a product in order to reduce dismantling costs and improve recyclability [46–48]. Furthermore, the variance should be kept low and easy access to connection points should be provided [42]. For the design of the new interfaces the CAD program Siemens NX was used.

## 4. Results

This chapter is divided into three sections. The first section analyses different interface concepts and describes the results of benchmarking. The second section illustrates the process steps of the design process. Subsequently, a systematic comparison of the new concept with the existing concept is carried out.

### 4.1. Benchmarking of Different Front Bumpers

As described in the previous chapter, the interface concepts of different electric vehicles are analyzed and compared by the following criteria: Interface type, interface diversity, interface accessibility, cable connections, and material diversity.

The interfaces of the front bumper of an Audi e-tron are evaluated below as an example (shown in Figure 4). The front bumper is attached to the vehicle mainly in the underbody area (3) using seven Torx screws. In the front area (1), the connection to the vehicle is achieved using two Torx screws. The side area (2) is also connected to the vehicle's fender using one Torx screw on each side (interface diversity: 5 points). The advantage of this design is that both the underbody and the fender linings of the vehicle do not have to be removed to dismantle the front bumper. In the side area (2), only the fender extension has to be dismantled to be able to loosen the screw connection underneath. For this reason, interface accessibility is assessed with three points. The non-destructive interface types are rated with five points. Assuming that all Torx screws can be opened with one tool, as well

as the fact that only the two fender flares need to be disassembled, this connection of the front bumper to the vehicle offers a way to quickly and easily disassemble and replace the front bumper. Dismantling is further simplified by a single cable connection (5 points) to the vehicle, whereas the material diversity of four different plastics makes the separation more difficult (material diversity: 2 points).

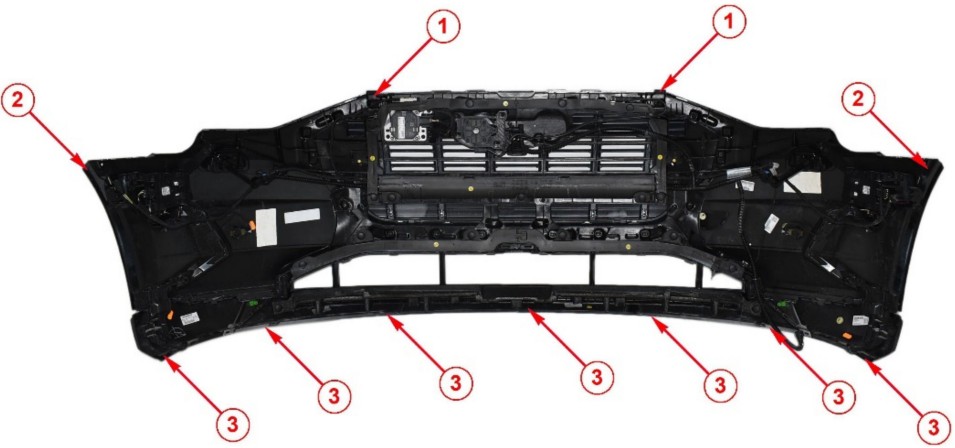

**Figure 4.** Interfaces of the front bumper of an Audi e-tron.

Figure 5 shows the benchmark results for all examined concepts graphically. The lack of interface accessibility results from the additional removal of neighboring components when dismantling the front bumper and is identified as the main problem for easy and fast dismantling. Since components such as the underbody and fender linings are mounted above the front bumper after it, current concepts of attachment do not allow fast and automated disassembly [49–56]. Another problem is the large number of cable connections that connect the sensors to the vehicle.

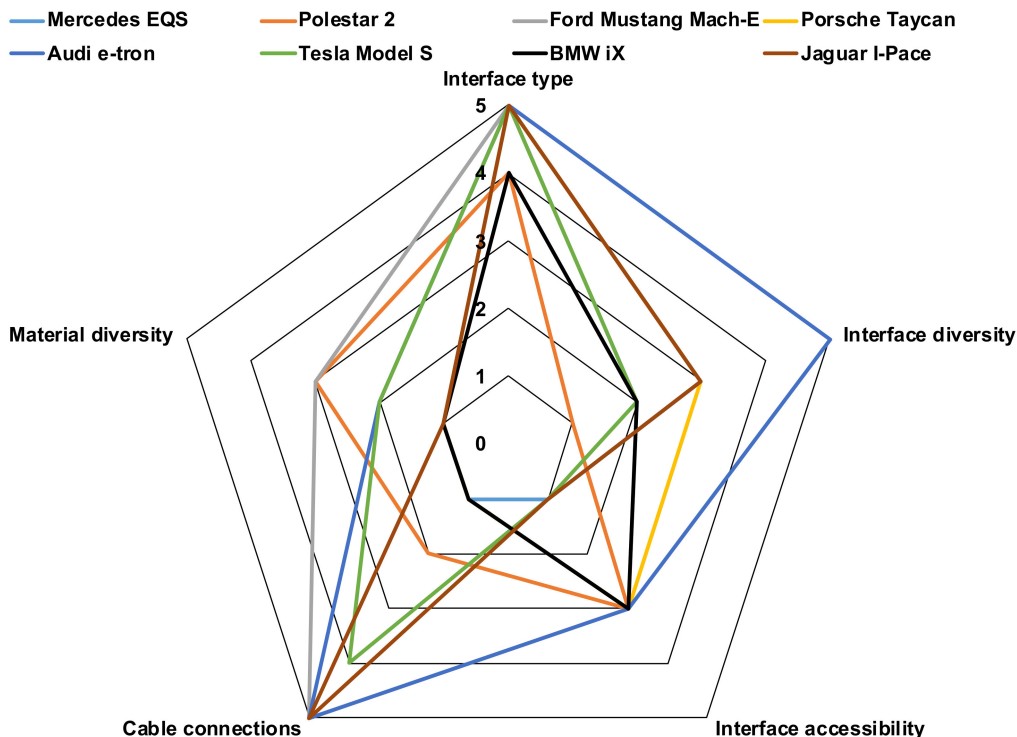

**Figure 5.** Comparison of the interfaces of different electric vehicles by defined criteria.

On the one hand, these connectors have to be joined by personnel effort during assembly and, accordingly, disconnected again during disassembly. Audi's concept, consisting

of non-destructive connections using Torx screws as well as easy access to the interfaces and a single cable connection, already currently offers the potential for a rapid dismantling of the front bumper [49]. However, as with other manufacturers, this concept uses a large number of screw connections and still requires the dismantling of an add-on part on each side. Another remarkable characteristic of the front bumper is the wide variety of materials used by most vehicle manufacturers [49,50,52,53,55,56]. This results in time-consuming dismantling and separation of the front bumper after it has been removed. On the positive side, most manufacturers have paid attention to the non-destructive dismantling of the front bumper. In some cases, a large number of different interfaces are used, such as various screw and snap-on connections. This necessitates the use of different tools and thus also results in a longer disassembly time. The long dismantling time is seen as the reason for the low proportion of dismantled components from end-of-life vehicles in dismantling operations. Moreover, the number of individual parts in the different front bumpers is also noticeable. Here, a range from 10 individual parts at Tesla to 33 individual parts at Mercedes-Benz can be determined [53,56]. On average, the front bumpers of the vehicles studied consist of around 19 individual parts. As the number of individual components increases, so does the number of plastics used in the front bumper. The front bumper contains an average of 5.4 different plastics. The range extends from three (Polestar, Ford) to nine (Mercedes-Benz) plastics used [51,53,54]. Therefore, it can be concluded that a reduced number of individual components goes hand in hand with a lower number of different plastics used.

*4.2. Design Process*

The design process is conducted in accordance with VDI guideline 2223 [44]. The aim is an easy and fast disassembly of the front bumper. One option is to install the front bumper both over the underbody, as some manufacturers have already done, and over the fender linings. However, this leads to complicated disassembly and reassembly, as well as a large number of connections being detached in the side area and on the underbody, which results in a longer disassembly time. For this reason, the solution principle of direct connection is not pursued further, and instead, the focus is placed on the insertion of an intermediate level, which is installed as a connecting part between the front bumper and the front module. The basic idea behind this solution principle is to decouple the neighboring components from the actual front bumper. As a result, the neighboring components of the front bumper are no longer directly connected to it but are mounted on the intermediate level.

Two alternatives can be distinguished for the solution principle of adding an intermediate level, which is inserted between the front bumper and the front module (see Figure 6): On the one hand, a one-piece solution and, on the other, a split solution consisting of two individual parts. Both alternatives were constructed and then evaluated for their advantages and disadvantages.

The requirements for pedestrian protection state that the flexible front bumper must feature a sufficient distance from the next rigid component behind it. Moreover, the requirement for a versatile design of the new interfaces is that the new interface concept must also be up-to-date in the future if the geometry of the front bumper is changed. In terms of implementing an intermediate level, however, this means that it cannot be flush with the front bumper. In addition, direct mapping of the front bumper contour is not permitted, as otherwise, the shape of the intermediate level will no longer be up to date in the event of future geometry changes.

The one-piece solution alternative is presented first (see Figure 6). Since the interfaces in the area of the dark-grey colored panel (2) may change with each new model or model update, they may not be part of the intermediate level. Due to strength reasons, bolted connections must be used in this area. A connection in the side area (1) via two webs to the front module, therefore, seems sensible. The aim is to decouple the front bumper from the neighboring components by integrating the intermediate level. At the same time,

however, it must be ensured that the overall stability of the front module is not impaired by the decoupling.

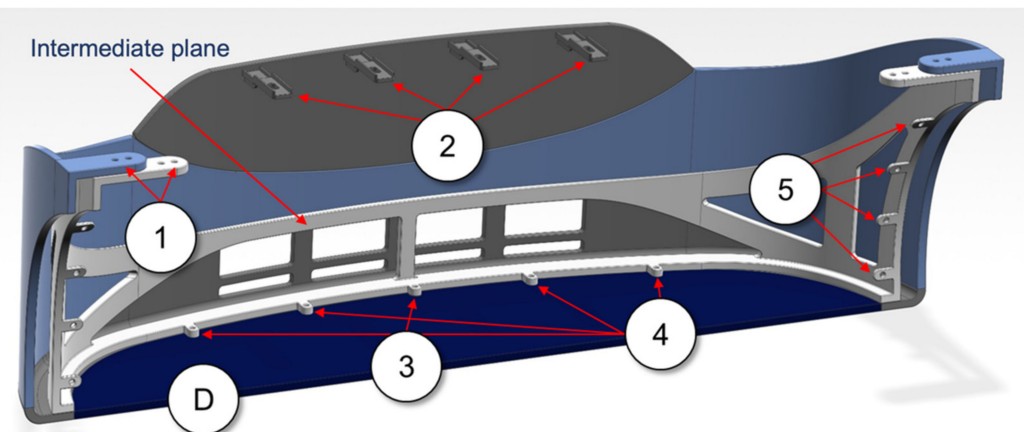

**Figure 6.** Front bumper with new one-piece interface concept.

As the connection of the front bumper to the front module remains in the area of the dark-grey colored panel (2), further connections to the intermediate level are required to achieve overall stability. This is accomplished by a connection in the center of the lower part of the front bumper, where it is connected to the intermediate level by a screw connection (3). Connections in the side area (1) are also possible. For this purpose, the front bumper is mounted over the intermediate level with two screw connections on each side. As shown in Figure 6, the intermediate level offers several connection options. On the underbody (D), for example, four further screw connections (4) can be identified in addition to a central screw connection of the front bumper (3). The underbody is bolted to the intermediate level via these connections. In the area of the fender linings (5), there are four screw connections on both sides. Thus, the front bumper is only indirectly connected to the neighboring components, as both the front bumper and the neighboring components are fastened to the intermediate level. Fixing the side area and the underbody ensures that the required distance between the front bumper and the intermediate level is maintained. In addition, the distance between these two components, as well as the chosen shape of the intermediate level, allow the geometry of the front bumper to be adjusted in the course of design changes without having to modify the intermediate level.

Another solution option is a two-part interface concept (see Figure 7). Here, the intermediate level is only inserted in the side area of the front bumper. In the front area (6), an additional connection to the front module behind is necessary for stabilization. This is done by using a screw connection. In contrast to the one-piece interface concept, in this solution, the underbody is not mounted on the intermediate level. Instead, the underbody is mounted in the rear part of the front module (7) and the front bumper, as already implemented by some manufacturers [49–51,54,55], is mounted over the underbody with five screw connections (8).

In terms of rapid disassembly, the one-piece solution alternative offers better conditions, as four fewer bolts need to be removed from the underbody compared with the two-piece solution variant. In addition, the stabilization of the underbody by the intermediate level facilitates the reassembly of the front bumper. During initial assembly, it is possible to mount the one-piece variant of the intermediate level together with the front bumper in one assembly step. The two-piece solution, on the other hand, is to be mounted separately from the front bumper, as it is additionally mounted to the front module at position (6) using screw connections. The front bumper could then be mounted on the intermediate level in a second assembly step. The main connection to the front module is still made via the radiator grille in the center area (2). If the interfaces change in the future, this does not affect the intermediate level. Only the connections in the side area (1) and on

the underbody (4) are taken as given and must not change accordingly in future models to continue to ensure compatibility between the front bumper and the intermediate level.

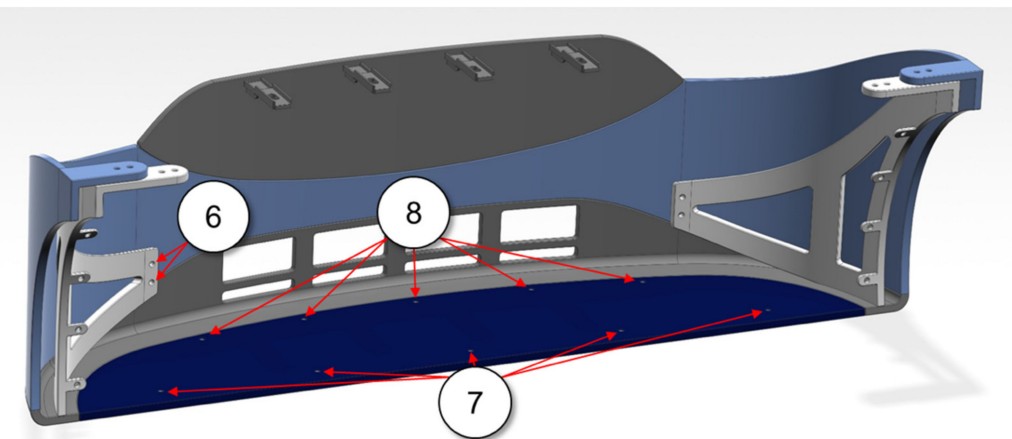

**Figure 7.** Front bumper with two-part interface concept.

### 4.3. Comparison of Disassembly Steps of the Current Solution vs. New Concept

The disassembly steps of the current interface solution are compared with the new interface concept. The disassembly map as shown in Figure 8 consists of various white circles, each of which represents a component of the overall product. All circles are interconnected by arrows, which indicate the disassembly direction, whereby the start is always at the assembled product. The type of disassembly movement depends on the tool required and is visually represented in the action blocks by different block shapes and colors. Connecting elements such as snap hooks, hinges, and cable plug connections can often be released with a hand movement. They are represented by a green rounded rectangle [57]. Connections that can be released with the single motion of a tool are represented by an orange rectangle. These tools include, for example, hammers or cutting pliers [57]. For joints with multiple movements, the action blocks are represented as a red hexagon shape. Examples include screw connections, which require multiple rotational movements to loosen the connection due to the use of a screwdriver [57]. The disassembly time depends on the actions required to disassemble a component. Action blocks can be used to specify the type of disassembly movement in the disassembly map (e.g., loosening screws). If the same type of fastener (e.g., screw) and the same tool is used (e.g., M8 screwdriver), the number of repetitions is indicated next to the action block. This provides an overview of the tools required and the number of connections to be loosened [57].

To disassemble the front bumper of a Mercedes EQS, several steps are required as shown in Figure 8. First, the hood of the vehicle must be opened and the battery disconnected in the front area. Then the radar sensor (shown as B in Figure 8), which is located behind the star in the radiator grille, is removed. Now some add-on parts adjacent to the front bumper must be removed. On the underside of the vehicle, the engine compartment trim, consisting of four individual components, must be removed (shown as C–E in Figure 8). Then the fender linings (shown as F in Figure 8) on both sides of the vehicle can be removed. After these attachments have been removed, the front bumper connections can be disconnected. In addition, the electronic connections must be disconnected. Now the front bumper can be removed from the vehicle by two people.

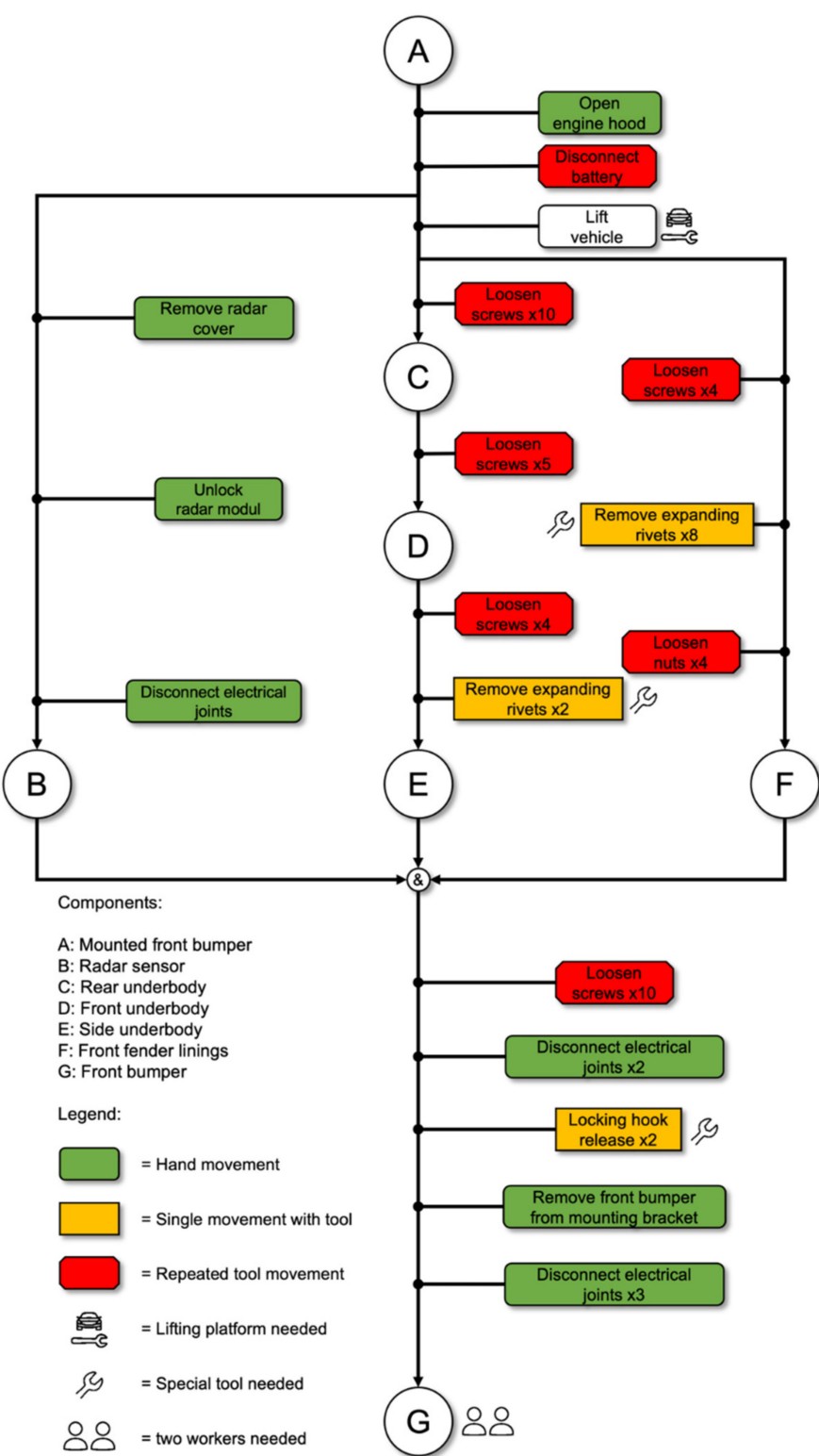

**Figure 8.** Disassembly steps of the current interface solution from the front bumper of a Mercedes EQS.

Figure 8 shows the disassembly steps of the front bumper of a Mercedes EQS, and the significant reduction in process steps for the new concept is shown compared to the current one, which is Figure 9. It should be noted that this is a strong simplification. It only focuses on the number of steps, but not on their complexity. The analysis, therefore, provides the first estimation.

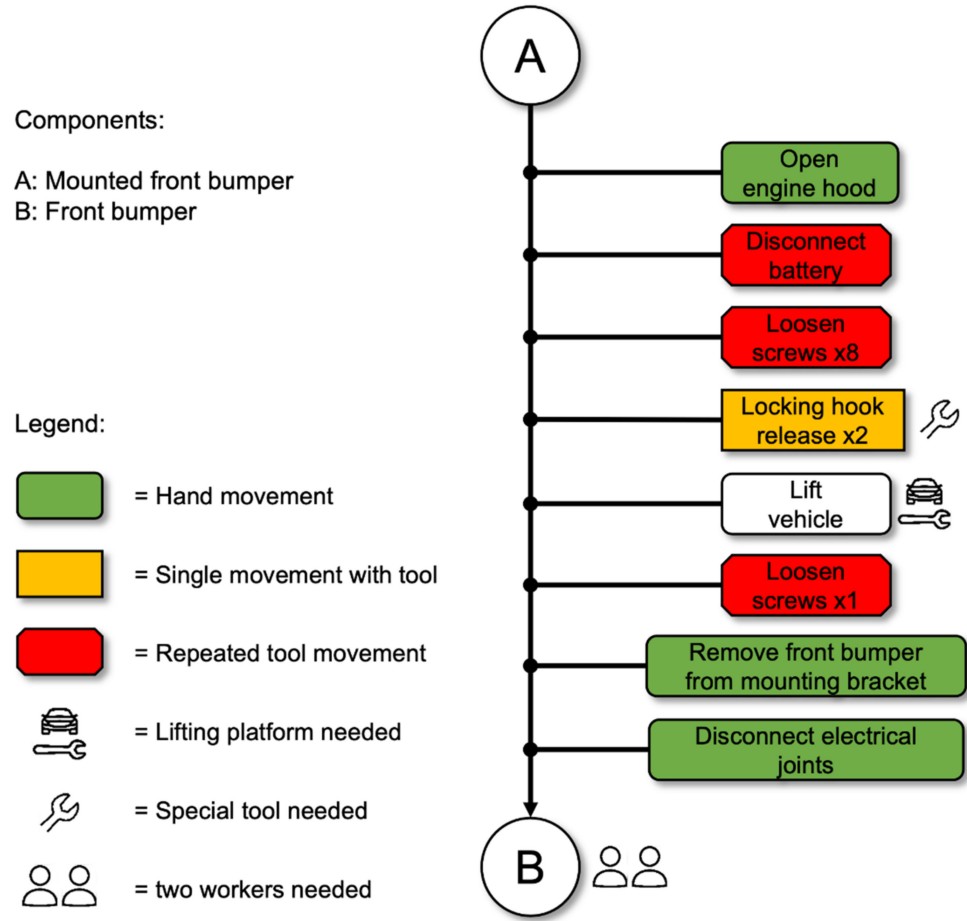

**Figure 9.** Disassembly steps of the front bumper with new interface concept.

The number of process steps is mainly reduced because fewer neighboring components need to be removed in the new concept for dismantling the front bumper. Consequently, the corresponding steps for loosening the screw connections, and expanding rivets and nuts, which are shown in Figure 8, are also eliminated. Apart from loosening the locking hooks, standardized screw connections result in only requiring one tool to loosen the connections. Summarizing, the new interface concept reduces the number of necessary process steps for dismantling from 24 to 9, which corresponds to a reduction of 62.5%. The two developed solution variants differ in four screws, which have to be loosened on the underbody in the two-part solution variant.

## 5. Discussion

The newly developed concept enables fast dismantling of the front bumper thanks to the easy accessibility of the interfaces. By reducing over 60% of the process steps required for dismantling the front bumper, the dismantling time is reduced by approximately 50% and at the same time, the way is opened for industrial dismantling. It should be noted that the new interfaces presented are at the concept stage. The reduction in process steps and the resulting shorter disassembly time has therefore only been proven theoretically and requires validation under real conditions. For this purpose, the new concept must be transferred to a real vehicle and its functionality, including crash safety and pedestrian protection, must be investigated. In addition, only bolted connections are used in the presented concept due to strength reasons. However, these have two disadvantages: firstly, bolted connections can be more expensive than other connecting elements, such as expanding rivets. Secondly, several hand movements are required when loosening the screw connections. This can lead to longer disassembly times compared to other fasteners such as snap hooks, which ideally

do not require tools for disassembly. As mentioned earlier, however, bolted connections must be used in some areas due to strength reasons.

The developed assessment method has some limitations. The criterion of interface diversity is based on the number of different types. The various types themselves are not evaluated concerning their respective necessary effort for loosening the connection. For example, snap hooks are more complex to loosen compared to standardized tools—especially with regard to automation.

Since the installation space of vehicles is severely limited and well utilized, it must also be examined which conceptual changes to the overall vehicle are necessary to be able to integrate the concept presented. Since all disassembly steps have been carried out manually up to now, it must also be investigated whether automated processes such as the explained human-robot collaboration can be implemented [40]. Companies are also increasingly turning to digitalization technologies such as artificial intelligence applications or digital twins to unlock circular economy potential [58,59]. These approaches can also enable automated disassembly in this context.

Furthermore, it is evident from the benchmarking analysis that the front bumper consists of a large number of different plastics and individual parts. This negatively influences the separation and recycling of the front bumper. Further investigations are therefore necessary to determine the extent to which the number of plastics and individual parts used can be reduced. The comparison between Tesla (10 individual parts) and Mercedes-Benz (33 individual parts) shows that a reduced number of individual components and thus a reduced number of different plastics used in the front bumper is possible [53,56]. A more detailed investigation of the reduction of the cable connections should also be aimed for.

Another factor that should not be neglected is the weight of the intermediate level. The additional component in the vehicle will increase the overall weight of the vehicle. One approach to solving this problem is to reduce the weight of the front bumper so that, in combination with the intermediate level, it has the same total weight as the front bumper as a whole. Nevertheless, increased weight caused by the intermediate level may lead to higher energy consumption and thus higher environmental impact in the use phase of the vehicle. On the other hand, more resources are extracted for the intermediate level which also increases the environmental impact. However, since reparability is improved thanks to easy dismantling, fewer resources are mined in general, and the energy required for dismantling is reduced. This may reduce the environmental impact both in the raw material extraction and in the production phase. This ecological advantage is to be compared with the increased environmental impact due to the weight increase of the intermediate level using life cycle assessments to determine the total ecological added value of the concept. In general, the contributions of a circular business model to environmental sustainability should be evaluated, especially in the case of technical innovations [31,60].

The concept of an intermediate level is also to be applied to other components that are frequently replaced. This may result in a vehicle that can be repaired and modernized without major expense. Future work should examine which components are suitable for this concept. The procedure for transferring the concept to other components will be similar to that described in this paper. The new interface concept, as well as the possibility of transferring this procedure to other components, represents an important contribution so that vehicles can be processed in an industrial dismantling plant and thus new business models can be implemented within the circular economy.

## 6. Summary and Conclusions

For the resource-intensive automotive industry, the circular economy offers great potential for decoupling sustainable economic growth from resource consumption. As part of new circular business models, vehicles will be loaned to customers in the future so that used vehicles return to the company after a defined period. In the production plant, the vehicles will be refreshed in terms of design or function, for example by replacing add-on parts. New technical innovations are needed to enable these parts to be replaced quickly.

In this work, new interface concepts were developed to ensure rapid and easy disassembly and subsequent reassembly of add-on parts. A methodology was developed to increase the disassembly of add-on parts. The methodology was applied to the front bumper as a component that is frequently damaged and needs to be replaced. The new interfaces should be designed in such a way that all the technical requirements for the front bumper are still maintained, and additional disassembly of neighboring components only has to take place to a minor extent.

In a benchmark, the current interface solutions implemented by various vehicle manufacturers were investigated, and potential weaknesses and strengths exist in these concepts with regard to fast and simple disassembly. The ease of disassembly of the front bumpers was evaluated according to defined criteria. Based on this benchmark, requirements were derived for the new concept to eliminate the weaknesses identified. Furthermore, two concepts for the new interfaces were developed. By inserting an intermediate level as a connecting part between the front bumper and the front module, the front bumper was decoupled from the remaining neighboring components. This results in the fact that no additional removal of neighboring components is necessary for dismantling the front bumper. Thanks to the new interface concept, the number of process steps required for dismantling the front bumper has been reduced by roughly 60% compared with the current interface solution. At the same time, the standardized screw connections mean that only one tool is required for disassembly. A significant reduction in disassembly time has thus been achieved and a basis for industrial disassembly has been provided.

Implementation of the new interface concept in the vehicle will involve extensive conceptual changes to the overall vehicle. Subsequent work will therefore need to investigate what measures are required to implement the new concepts in the vehicle. In addition, the theoretical concept presented must be transferred to a real vehicle to validate its functionality. Since only recommendations for a disassembly system were given in this work, it must be examined in what form such a system can be realized. Finally, analyses are required to reduce the variations of plastics and individual components used in a front bumper. This results in a significantly simplified separation of materials and thus in easier recycling of the front bumper. In the long term, the transfer to other components will result in a vehicle that can be repaired and modernized in a rapid and resource-efficient way. The new interface concept thus makes a significant contribution to the establishment of new business models within the circular economy and therefore achieves positive ecological and economic effects.

**Author Contributions:** Conceptualization, P.K., Y.D.D. and D.B.; methodology, P.K., D.B., Y.D.D. and S.K.; software, P.K.; formal analysis, P.K. and Y.D.D.; writing—original draft preparation, P.K. and Y.D.D.; writing—review and editing, Y.D.D., D.B., S.K. and R.M.; visualization, P.K.; supervision, R.M. All authors have read and agreed to the published version of the manuscript.

**Funding:** This research received no external funding.

**Institutional Review Board Statement:** Not applicable.

**Informed Consent Statement:** Not applicable.

**Data Availability Statement:** Not applicable.

**Conflicts of Interest:** The authors declare no conflict of interest.

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
