# Peer review of "Designing Component Interfaces for the Circular Economy—A Case Study for Product-As-A-Service Business Models in the Automotive Industry"

_sustainability, doi:10.3390/su142113851_

Round 1
Reviewer 1 Report
The article is interesting and generally, it deserves to be published with some revisions that are suggested below:
1. In the Abstract you need to describe in short, there two concepts for the new interfaces that were developed; inserting an intermediate level as a connecting part between the front bumper and the front module, the front bumper decoupled from the remaining neighboring components, and the standardized screw connections mean that only one tool is required for disassembly.
2. Would you please state how to validate the performance between the traditional way and the new interface concepts that were developed for the front bumper was selected as a component that is frequently damaged and needs to be replaced, how many times were tested, how many people were tested, tested the same type and same the brand vehicle?
3. Can you show us how and what steps to reduce by roughly 60% compared with a current interface solution when adapting the new interface concept, the number of process steps required for dismantling the front bumper?
Author Response
Dear Reviewer 1,
Thank you very much for your extensive comments. Please see attachment for our responses.
Kind regards,
Philip Krummeck, Yagmur Damla Dokur, Daniel Braun, Steffen Kiemel, Robert Miehe

Reviewer 2 Report
The topic of this research is related to the theme of this journal. Please find enclosed my constructive remarks and suggestions, for your guidance.
The introduction provided a good background to this manuscript. However, the authors are expected to clarify their research questions and the contribution of this paper.
The authors should substantiate their statements by using good references. They may include more references from this journal and from other high impact sources. I recommend the inclusion of the following:
Camilleri, M. A. (2020). European environment policy for the circular economy: Implications for business and industry stakeholders. Sustainable Development, 28(6), 1804-1812.
Corvellec, H., Stowell, A. F., & Johansson, N. (2022). Critiques of the circular economy. Journal of Industrial Ecology, 26(2), 421-432.
Kirchherr, J., Reike, D., & Hekkert, M. (2017). Conceptualizing the circular economy: An analysis of 114 definitions. Resources, conservation and recycling, 127, 221-232.
The more references you have, the better the quality of your paper. Academic authors are expected to build on extant knowledge and to appraise the contributions of their colleagues.
The researchers could have discussed further about the interfaces of different electric vehicles (by defined criteria). It is very difficult for the readers of this paper to compare among various electric vehicles. E.g. The shapes of Audi e-tron and of Porche Taycan are not clear.
Moreover, the authors should ensure that they are clearly explaining each figure (for the benefit of the readers of this paper, who may be non technical experts).
In conclusion, the researchers discussed about the implications to practitioners. They should elaborate about the theoretical implications of this case study. The authors should also identify the limitations of their research and outline future research directions to academia.
I look forward to reading your revised manuscript.
Best wishes.
Author Response
Dear Reviewer,
Thank you very much for your extensive comments. Please see the attachment for our responses.
Kind regards,
Philip Krummeck, Yagmur Damla Dokur, Daniel Braun, Steffen Kiemel, Robert Miehe

Round 2
Reviewer 2 Report
I noticed that you refined many areas in your manuscript. I am recommending the acceptance of your paper. Well done and congratulations.
P.S. You may consider reading and citing this article on product service systems: Camilleri, M. A. (2019). The circular economy's closed loop and product service systems for sustainable development: A review and appraisal. Sustainable Development, 27(3), 530-536.
Best wishes.